# Transcriptomic Analysis of Human Skeletal Muscle in Response to Aerobic Exercise and Protein Intake

**DOI:** 10.3390/nu15153485

**Published:** 2023-08-07

**Authors:** Xueqing Zeng, Linghong Li, Zhilin Xia, Lianhong Zou, Timothy Kwok, Yi Su

**Affiliations:** 1Key Laboratory of Molecular Epidemiology of Hunan Province, School of Medicine, Hunan Normal University, Changsha 410013, Chinaxiazhilin@hunnu.edu.cn (Z.X.); 2Hunan Provincial Institute of Emergency Medicine, Hunan Provincial People’s Hospital, Changsha 410009, China; 3Department of Medicine & Therapeutics, Prince of Wales Hospital, The Chinese University of Hong Kong, Hong Kong SAR, China

**Keywords:** skeletal muscle, aerobic exercise, protein intake, transcriptomic analysis

## Abstract

This study aimed to provide a more comprehensive molecular insight into the effects of aerobic exercise (AE), protein intake (PI), and AE combined with PI on human skeletal muscle by comparing their transcriptomic profiles. Fourteen published datasets obtained from the Gene Expression Omnibus (GEO) database were used. The hub genes were identified in response to acute AE (*ACTB*, *IL6*), training AE (*UBB*, *COL1A1*), PI (*EZH2*), acute AE combined with PI (*DDIT3*), and training AE combined with PI (*MYC*). Both *FOS* and *MYC* were upregulated in response to acute AE, and they were, respectively, downregulated by higher PI and a combination of AE and PI. *COL1A1* was upregulated by training AE but was downregulated by higher PI. Results from the gene set enrichment analysis (*p* < 0.05 and FDR < 25%) showed that AE and PI delivered their impacts on human skeletal muscle in analogous pathways, including aerobic respiration, mitochondrial complexes, extracellular matrix (ECM) remodeling, metabolic process, and immune/inflammatory responses, whereas, PI may attenuate the response of immune/inflammation and ECM remodeling which would be promoted by AE, irrespective of its types. Compared to PI alone, acute AE combined with PI would further promote protein turnover and synthesis, but suppress skeletal muscle contraction and movement.

## 1. Introduction

Muscle metabolism plays an essential role in the genesis of many pathologic conditions and chronic diseases [1]. Sarcopenia is an aging-related loss of muscle mass and/or muscle function [2], which involves adverse muscle changes across the lifetime and the increased risk of falls, fractures, frailty, and mortality [3]. The prevalence of sarcopenia and muscle atrophy is up to 29% in community-dwelling populations and 33% in long-term care populations [4], which results in a large healthcare burden [5]. The development of optimal interventions to maintain muscle and improve mobility is extremely important.

Exercise could help preclude the pathophysiological processes of muscle mass and function, and prevent or delay the development of sarcopenia [6,7,8]. Considerable evidence demonstrates that resistance exercise (RE) is the most effective in enhancing muscular strength and function performance [9,10], and RE is now recommended as the first-line non-pharmacological intervention for counteracting the deleterious consequences of sarcopenia [11]. However, besides the loss of muscle mass, the decrease in muscle mitochondria with age-related muscle atrophy [12,13] contributes significantly to an imbalance of energy homeostasis, and the development of sarcopenia [14,15,16]. Chronic conditions, such as cardiovascular diseases, may accelerate the muscle dysfunction and tissue atrophy through imparting pathological hormonal change, malnutrition, and physical inactivity [17,18], and obesity may contribute to mitochondrial dysfunction through the intramuscular accumulation of impaired β-oxidation capacity [19]. Differing from the exercise modality of RE, aerobic exercise (AE) is more effective in improving aerobic capacity, maintaining whole-body energy homeostasis [9,10,20], managing the metabolism of visceral adipose tissue in obesity [20,21], and improving cardiovascular fitness through the increase in exercise capacity and peak oxygen uptake [22,23], which would benefit muscle health in a divergent pathway. Moreover, previous studies found that AE imbued to mid- and long-term immune modulations [24] and induced strong cellular immune responses and more alternations in immune cell populations [25]. Compared to RE alone, AE appears to be more effective for maintaining and improving maximum aerobic power. Moreover, mixed training or RE in combination with AE was suggested to deliver more positive effects on muscle mass, strength, and function in older people [26,27].

Nutritional interventions have also been recommended as a non-negligible option to preserve muscle mass and function [28,29,30,31]. Double-blind RCT suggested that higher protein intake (PI) brought more benefits than lower PI in preventing sarcopenia in older people [32]. The synergistic effect of exercise and PI has been identified [33]. Besides the positive effect of the combination of RE and PI on skeletal muscle [31,34,35,36], the potential benefit brought by the combination of AE and protein feeding on skeletal muscle myofibrillar plasticity [37,38], greater gains in VO_2max_, and stimulation of lean mass accretion have also been reported [9,39,40,41]. However, controversies still exist over the capability of the combination of AE and dietary protein on improving skeletal muscle oxidative capacity and endurance performance [42,43,44,45].

More evidence is needed to identify whether AE or and its combination with PI could also be effective interventions for improving human skeletal muscle health. Understanding the key molecular signature characteristics underlying responses to these interventions may bring some new hints. Transcriptomic analysis can provide a unique microscopic insight to access the directions of adaptive changes and predict the underlying mechanisms. Previous studies have used transcriptomic analysis to summarize the overlap and the difference of human skeletal muscle in response to RE, AE, acute exercise, training exercise, and inactivity/disuse [39,40,41]. Nuclear Receptor Subfamily 4 Group A Member 3 (*NR4A3*) and SMAD Family Member 3 (*SMAD3*) were identified as two of the most exercise- and inactivity- and acute- and training-responsive genes, respectively [39,41]. The transcriptional signatures of disuse and RE training (RET) suggested that RET was not completely effective in countering disuse atrophy due to the fact that the mechanisms of disuse were not covered by the converse of the RET [40].

Thus, the present study aimed to highlight the divergent transcriptomic responses of human skeletal muscle to AE, PI and their combinations, and to provide more molecular evidence of these interventions for maintaining muscle health and alleviating sarcopenia.

## 2. Materials and Methods

### 2.1. Public Datasets

The Gene Expression Omnibus (GEO) database [46] was searched (up to 1 December 2022) to obtain datasets of the gene expression profiling on human skeletal muscle in response to AE, PI, and AE combined with PI. Inclusion criteria were as follows: (1) datasets using microarray were chosen; (2) studies conducted in healthy adults (i.e., free of injury and disease); (3) human muscle biopsy samples were used; and (4) studies with 5 subjects or more were included. If there was more than one record of time points for muscle biopsy, we selected the one with the minimum difference across the studies to improve comparability. The studies, which had results reported separately for gender or reported separately for different age groups, were treated as independent studies.

### 2.2. Data Pre-Processing

To minimize the heterogeneity across the multiple microarray platforms, quantile normalization and log 2 transformations were used for each microarray expression matrix data. If more than one probe was used for the gene detection, the probe with the maximal average expression would be taken. Whereas, if a probe was used for more than one gene, the probe would be discarded due to its low specificity. If multiple categories were used to assess the degree of PI, the category with the highest degree would be chosen to compare with the one with the lowest degree.

### 2.3. Identification of Differentially Expressed Genes (DEGs)

Differential expression analysis (DEA) was performed using the Benjamini–Hochberg (BH) correction for paired samples using the limma package in R (version 4.2.0) [47]. The genes with *p* value less than 0.05 in DEA were identified as the DEGs. These DEGs with a threshold of log2 Fold Change (log2FC) above 0 were considered to be upregulated, while those with log2FC less than 0 were downregulated.

### 2.4. Functional Analysis and Network Visualization of co-DEGs

Those DEGs which were identified in two or more datasets were labeled as co-differentially expressed genes (co-DEGs). Gene Ontology (GO) and the Kyoto Encyclopedia of Genes and Genomes (KEGG) pathway enrichment analysis were performed to identify the biological functions and potential pathways of the co-DEGs. The bioconductor package “org.Hs.eg.db” and the “cluster profiler” package [48] were used for GO and KEGG pathway enrichment analysis.

Subsequently, all the co-DEGs were used to construct a gene expression network to find the interactions among these genes using Search Tool for the Retrieval of Interacting Genes/Proteins (STRING) Database (https://cn.string-db.org/, accessed on 26 December 2022, version: 11.5). The gene with the top degree value calculated by CytoHubba plugin [49] in Cytoscape software (version 3.9.1) was considered the hub gene (i.e., the highly connected genes within a module and those significantly associated with biological function).

### 2.5. Gene Set Enrichment Analysis (GSEA)

GSEA among different interventions was further performed to explore genome-wide expression profiles within each dataset, without setting up an arbitrary threshold. This would provide an alternative way to screen out obvious differential biological functions induced by varied interventions within each dataset. GSEA was performed using the “cluster profiler” package. The number of permutations was set up at 10,000. Gene set was considered to be significantly enriched when the absolute value of normalized enrichment score (NES) was more than 1 with a *p*-value less than 0.05 and a false discovery rate (FDR) less than 25%.

## 3. Results

### 3.1. Overview of Datasets Collection

There were fourteen datasets included in the present study (Appendix A). Those were studies of acute AE (GSE43856 [50], GSE59088 [51], GSE126296 [52], GSE59363 [53]), training AE (GSE111551, GSE72462 [54], GSE1786 [55], GSE43760 [56]), PI (GSE8441 [57], GSE9419 [58], GSE73525 [59]), acute AE combined with PI (GSE27285 [60], GSE44818 [61]), and training AE combined with PI (GSE147494 [62]). Acute AE refers to a bout of aerobic exercise, and training AE refers to regular aerobic exercise with frequency and intensity lasting for at least 2 weeks. Both GSE8441 and GSE72462 can be considered to be two independent studies by gender, GSE9419 can be considered two independent studies by age group (the younger one (<50 years) and the older one (≥50 years)). The details of included datasets were shown in Table 1.

### 3.2. DEGs Identification, Hub Gene Screening, and Gene Expression Network Construction

The details of the number of DEGs from each dataset and co-DEGs among different populations across interventions were summarized (Figure 1 and Appendix A). Compared to PI or AE combined with PI, more DEGs were identified in the studies delivering AE alone (Figure 1A). No co-DEG were reported for training AE, PI, and training AE combined with PI in the younger male (YM) due to only one dataset for each of them being included. Herein, the DEGs were considered the co-DEGs in these studies (Figure 1B).

The top five up- and down-regulated DEGs within each dataset and co-DEGs within each intervention among different populations in response to varied interventions were summarized (Appendix A). Nuclear Receptor Subfamily 4 Group A Member 3 (*NR4A3*), Pyruvate Dehydrogenase Kinase 4 (*PDK4*), PPARG Coactivator 1 Alpha (*PPARGC1A*) were consistently upregulated in response to acute AE (Appendix A). These genes were reported to contribute to the regulation of glucose metabolism, cell proliferation, differentiation and transformation, and the transcription by RNA polymerase II. MicroRNA 206 (*MIR206*), and Circadian-Associated Repressor of Transcription (*CIART*) were the most downregulated co-DEG in response to acute AE in the YM and the OM, respectively (Appendix A).

Stearoyl-CoA Desaturase (*SCD*), Rho GTPase Activating Protein 12 (*ARHGAP12*), and Asporin (*ASPN*) were the most upregulated genes in response to training AE in the YM, the OM, and the older female (OF), respectively (Appendix A). Ankyrin Repeat Domain 2 (*ANKRD2*), G Protein-Coupled Receptor 37 (*GPR37*), and Establishment of Sister Chromatid Cohesion *N*-Acetyltransferase 1 (*ESCO1*) were the most downregulated co-DEGs in response to training AE in the the YM, the OM, and the OF, respectively (Appendix A).

Chromosome 10 Open Reading Frame 126 (*C10orf126*), Secretoglobin Family 2A Member 2 (*SCGB2A2*), and EPH Receptor A2 (*EPHA2*) were the most upregulated co-DEGs in response to PI in the YM, the OM, and the OF, respectively (Appendix A). Splicing Factor Proline/Glutamine-Rich (*LOC654780*), Adhesion G Protein-Coupled Receptor G2 (*ADGRG2*), and Mannose Binding Lectin 2 (*MBL2*) were the most downregulated genes in response to PI in the three groups, respectively (Appendix A). Noteworthy, *FOS* was significantly downregulated in response to PI but was upregulated in response to acute AE.

Cell production-, proliferation-, and differentiation-related genes, including Interferon Related Developmental Regulator 1 (*IFRD1*) and DNA Damage Inducible Transcript 3 (*DDIT3*), were consistently upregulated in response to acute AE combined with PI. Protein Phosphatase 1 Regulatory Subunit 15A (*PPP1R15A*) was the most downregulated co-DEG in response to acute AE combined with PI. What is noteworthy is that some immune/inflammation-related genes such as C-C Motif Chemokine Ligand 2 (*CCL2*) and C-X-C Motif Chemokine Receptor 7 (*CXCR7*) were downregulated in response to acute AE combined with PI (Appendix A). Genes involved in ECM remodeling, including Collagen Type I Alpha 1 Chain (*COL1A1*) and Collagen Type III Alpha 1 Chain (*COL3A1*), were downregulated both in response to PI and acute AE combined with PI (Appendix A). Similar to PI alone, *FOS* was downregulated in response to acute AE combined with PI. Most of the top five responded genes to PI and acute AE combined with PI were strongly associated with the immune/inflammation process and ECM remodeling.

The top upregulated and downregulated genes in response to training AE combined with PI were CCR4-NOT Transcription Complex Subunit 7 (*CNOT7*) and Solute Carrier Family 13 Member 2 (*SLC13A2*), respectively. They were reported to be related to the negative regulation of cell proliferation and the formation of kidney stones, respectively (Appendix A).

A minimum interaction score of 0.4 was set up for the gene expression network construction in the STRING database using the identified co-DEGs. The degree value was calculated and the top five genes were summarized for each network (Appendix A). Actin Beta (*ACTB*) and Interleukin 6 (*IL6*) were the hub genes in response to acute AE in the YM and the OM, respectively. Ubiquitin B (UBB) and *COL1A1* were the hub genes in response to training AE in the YM and the OF, respectively. The enhancer Of Zeste 2 Polycomb Repressive Complex 2 Subunit (*EZH2*), *DDIT3*, and MYC Proto-Oncogene (*MYC*) were the hub genes in response to PI, acute AE combined with PI, and training AE combined with PI in the YM, respectively (Appendix A).

*FOS*, *MYC*, and *COL1A1* identified above were upregulated by acute AE and training AE, but were downregulated by PI and acute AE combined with PI (Appendix A). Subsequently, the expression levels of these genes in the intervention and control groups within each dataset were compared across the different interventions. Compared to each own control group, the expression levels of *FOS*, *MYC*, and *COL1A1* were higher in acute AE and training AE but lower in the PI-related studies, although statistical significance was not shown in all comparisons (Appendix A).

### 3.3. Functional Enrichment Analysis for co-DEGs

GO terms and KEGG pathways enrichment analysis were conducted to explore the biological functions of the co-DEGs. For acute AE studies, co-DEGs were mainly enriched in the terms related to starvation, peptide hormones, protein turnover and synthesis, cellular response to external stimuli, and the circadian regulation of gene expression (Figure 2A and Appendix A). The AGE-RAGE signaling pathway in diabetic complications and circadian rhythm was the most significant KEGG pathway in response to acute AE in the YM and the OM, respectively (Figure 3A and Appendix A). For training AE, the most significant GO terms in the YM, OM, and OF were fatty acid transmembrane transport, fatty acid beta-oxidation using acyl-CoA dehydrogenase, and positive regulation of smooth muscle cell proliferation, respectively (Figure 2B and Appendix A, Appendix A). No significant KEGG pathway was enriched in response to training AE (Figure 3B and Appendix A).

For PI in the OM, the most significant GO term was the positive regulation of vasoconstriction (Appendix A and Appendix A). No GO term was significantly enriched in the co-DEGs of PI after BH correction in the YM and OF (Figure 2C and Appendix A, Appendix A). And, no KEGG pathway was significantly enriched by co-DEGs in response to PI (Figure 3C and Appendix A, Appendix A). The co-DEGs of acute AE combined with PI were mainly associated with the function or pathway of protein synthesis and turnover, such as endoplasmic reticulum unfolded protein response and cellular response to unfolded protein. This was similar to acute AE but not to PI alone in the YM (Figure 2D and Figure 3D, Appendix A). Few GO terms and KEGG pathways were significantly enriched by the co-DEGs from the response to training AE combined with PI (Figure 2E and Figure 3E, Appendix A).

### 3.4. GSEA

Consistent results were shown across the studies of acute AE. The gene sets which were significantly upregulated in response to acute AE included the cellular response to external stimuli, regulation of fatty acid oxidation, energy metabolic process, ECM remodeling, vasculature development, skeletal muscle development, and immune/inflammation process (Figure 4 and Appendix A, Appendix A).

Compared to acute AE, more profound transcriptomic changes in the ECM remodeling and immune/inflammation process were observed in response to training AE in the older people. Gene sets related to extracellular stimuli, protein turnover and synthesis, and angiogenesis regulation were not significantly enriched in response to training AE (Appendix A). Compared to the older people, mitochondrion complex, aerobic respiration, electron transport complex, and ATP synthesis were downregulated in the YM (which were upregulated in the older people) (Appendix A). Gene sets related to ECM organization, basement membrane, cornified envelope, and peptide cross-linking were upregulated in response to training AE in the OM, while nuclear glucocorticoid receptor binding, nuclear receptor activity, and ligand-activated transcription factor activity were downregulated (Appendix A). In the OF, the inflammation/immunity-related gene sets were profoundly upregulated in response to training AE, while gene sets related to nuclear glucocorticoid receptor binding, keratin filament, and intermediate filament were downregulated (Appendix A).

Gene sets in relation to aerobic respiration, mitochondrial complex, and electron transport complex were upregulated in response to PI in the older people (Appendix A). Notably, gene sets in relation to the immune/inflammation process, collagen fibril organization, and ECM remodeling were significantly downregulated in response to PI (Appendix A), which were upregulated in response to AE.

Contrary to AE alone, gene sets associated with the immune/inflammation process, ECM remodeling, cellular response, muscle organ development, renal system development, and regulation of vasculature development were downregulated in response to acute AE combined with PI, which were opposite to their responses to AE alone (Appendix A). No gene set was significantly enriched in response to training AE combined with PI (*p* > 0.05, FDR > 25%, Appendix A).

KEGG gene sets’ enrichment analysis revealed that the oxidative phosphorylation pathway was consistently upregulated both in response to acute AE and PI (Appendix A). Pathways of mineral absorption, non-alcoholic fatty liver disease, ECM receptor interaction, immune-related signaling pathways, and cardiac muscle contraction were upregulated in response to acute AE (Figure 5 and Appendix A, Appendix A). The Fanconi-anemia-pathway-, spliceosome-, and Herpes-simplex-virus-1-infection-related gene sets were downregulated in response to acute AE in the YM, and the Wnt signaling pathway and long-term potentiation were significantly downregulated in the OM (Appendix A). Oxidative phosphorylation, diabetic cardiomyopathy, and cardiac muscle contraction gene sets were downregulated in response to training AE in the YM (Figure 5D and Appendix A). ECM-receptor interaction and malaria pathways were upregulated in response to training AE in the older people (Appendix A, Appendix A). The neuroactive ligand–receptor interaction pathway was upregulated, while no KEGG pathway-related gene set was significantly enriched in response to PI in the YM (Appendix A).

Gene sets concerning the olfactory transduction pathway were upregulated both in response to PI and acute AE combined with PI (Appendix A). Gene sets in the pathway of antigen processing and presentation, staphylococcus aureus infection, and phagosome were downregulated in response to PI in the OM (Appendix A). Herpes simplex virus 1 infection, systemic lupus erythematosus, and phagosome were downregulated in response to PI in the OF (Appendix A). Protein-digestion-and-absorption-, systemic-lupus-erythematosus-, and TNF-signaling-pathway-related gene sets were downregulated in response to acute AE combined with PI (Appendix A). Contrary to the response to acute AE, gene sets in AGE-RAGE signaling pathway (diabetic complications and cardiac muscle contraction) and ECM-receptor interaction pathway were downregulated to acute AE combined with PI (Appendix A). Additionally, no gene set in KEGG pathway was significantly enriched in response to training AE combined with PI (*p* > 0.05, FDR > 25%, Appendix A).

## 4. Discussion

The transcriptional profiles of human skeletal muscle in response to AE, PI, and AE combined with PI were analyzed in the present study. *ACTB*, *IL6*, *UBB*, *COL1A1*, *EZH2*, *DDIT3*, *MYC* was the hub gene in response to acute AE in the YM and the OM, training AE in the YM and the OF, PI in the YM, acute AE combined with PI in the YM, and training AE combined with PI in the YM, respectively. *FOS* was upregulated in response to acute AE but was downregulated in response to PI and acute AE combined with PI. *MYC* was upregulated in response to acute AE while it was downregulated in response to AE combined with PI. *COL1A1* was upregulated in response to training AE while it was downregulated in response to PI.

AE and PI may deliver impacts on human skeletal muscle in several analogous pathways, which included aerobic respiration, mitochondrial complexes, ECM remodeling, energy metabolic process, and immune/inflammatory responses. However, compared to PI, AE have an opposite impact on ECM remodeling and immune/inflammation process. Compared to PI, acute AE combined with PI may have additional impact on upregulating protein turnover and synthesis and downregulating musculoskeletal movement and muscle contraction. Few gene set was enriched in response to training AE combined with PI, which suggested that the limited observed biological impact on skeletal muscle induced by additional PI based on training AE.

### 4.1. Transcriptomic Changes in Skeletal Muscle in Response to Acute AE, PI, and Acute AE Combined with PI

The shared biological pathways and the overlap genes identified to acute AE and PI implied the molecular responses of muscle to them were not separate. The upregulated gene sets with both acute AE and PI alone were related to ATP synthesis and energy metabolic. AE could improve body metabolism by optimizing global energy expenditure and activating multiple neuroendocrine pathways [63], while more energy is needed to maintain skeletal muscle contraction due to rapid increase in mitochondrial oxidative enzymes during AE [64]. Meanwhile, high-protein diets could induce higher energy expenditure to regulate body metabolic status [65]. In the present study, mitochondrial-complex-related gene sets were consistently upregulated in acute AE, PI, and acute AE combined with PI, which suggested that both acute AE and PI may promote the aerobic respiration function. Moreover, AE could remodel the quantity and quality (function per unit) of skeletal muscle mitochondria to promote substrate oxidation and increase mitochondrial oxidative enzymes [64], while dietary protein-derived amino acid could incorporate into mitochondrial protein to promote mitochondrial content [45]. These may partly explain the feeble impact on mitochondrial content induced by acute AE, PI, and acute AE combined with PI.

The downregulation of immune/inflammation molecular process was identified both in response to PI and acute AE combined with PI in the present study, which was consistent with the reports in the previous RCTs [57,66]. Although inflammation induced by AE plays an important role in the increase in skeletal muscle mass and fiber components by skeletal muscle releasing cytokines, prostaglandins, and chemokines [67,68], PI may weaken the inflammation response. For example, the anti-oxidative effect of dietary whey protein may counteract chronic inflammation through regulation of the nuclear transcription factor kB signaling pathway and cytokine production [69]. ECM remodeling is also influenced by mesenchymal cells, and is closely related to the immune/inflammation process [70]. Reasonably, the response of ECM remodeling-related genes (sets) across acute AE, PI, and acute AE combined with PI was similar to the response of immune/inflammation in the present study.

The transcriptomic profile in response to acute AE combined with PI was comparable to PI alone, which may be due to the fact that the impact induced by acute AE had been offset in the combined intervention studies, where acute AE was always assigned to the control group. Meanwhile, the downregulation of skeletal muscle contraction, musculoskeletal movement, and muscle system development were found in response to acute AE combined with PI, which was contrary to the responses to AE alone. Thus, the additional impact may be induced by the combination of PI and AE. Postexercise PI, which may provide the most efficient gains in muscle plasticity and support myofibrillar protein synthesis as key components in defining contractile performance in response to AE [71]. Protein synthesis and turnover were upregulated both in response to acute AE and the combination of acute AE and PI but not in response to PI alone, which indicated that acute AE had a strong effect on protein synthesis and turnover and this effect may be enhanced when PI was added. This is reasonable in that muscle contraction during exercise, whether RE or EE, has a profound effect on muscle protein turnover [72]. These findings were also consistent with the reports from previous RCTs [73,74], which indicated that PI alone could not modulate myofibrillar protein synthesis (MyoPS) rates that were caused by disuse, while pre-sleep protein ingestion could stimulate myofibrillar and mitochondrial protein synthesis during an overnight recovery from AE. Protein ingestion may facilitate skeletal muscle conditioning following endurance training and may help athletes to improve endurance training efficiency [74]. Moreover, the type, dose, and time of PI may also contribute to the difference [75]. Protein supplementation studies suggested that a relatively higher protein intake was required to maximally stimulate skeletal muscle protein synthesis in the older people [76,77]. Greater PI intake during recovery from AE was important in the regulation of whole-body and muscle protein synthesis rates [45]. In addition, compared to plant protein, animal protein was more likely to be a stimulator of muscle protein synthesis [78].

*ACTB* are highly conserved proteins that are involved in cell motility, structure, integrity, and intercellular signaling, which helps organize and maintain the cellular morphology by virtue of facilitating the processes of migration, division, growth, signaling and shaping the cytoskeleton [79]. Meanwhile, previous studies found that *ACTB* might participate in the mechanism of vascular remodeling and thrombogenesis [80]. As the hub gene in response to acute AE in the OM, *IL6* encodes a cytokine that functions in inflammation, maturation of B cells, tissue regeneration, and metabolism, which was consistent with the upregulation of the inflammation/immunity process in response to acute AE. *FOS* plays an important role in the gene interaction network of muscle in response to acute AE both in the YM and the OM, and it was the gene which was most upregulated in response to acute AE but was downregulated in response to PI and acute AE combined with PI. As a regulator of cell proliferation, differentiation, and transformation, *FOS* was involved in multiple inflammation pathways [81], which was also suggested by the results of GSEA in the present study. As the hub gene identified in response to acute AE combined with PI in the YM, *DDIT3* is implicated in adipogenesis and erythropoiesis, is activated by endoplasmic reticulum stress, promotes apoptosis, and plays a crucial role in endoplasmic reticulum stress-induced apoptosis and autophagy, host innate immunity repression, and viral infection facilitation [82].

### 4.2. Transcriptomic Changes in Skeletal Muscle in Response to Training AE, PI, and Training AE Combined with PI

Different from the response to acute AE, a cluster of ATP-metabolic-processes-, oxidative-phosphorylation-, aerobic-respiratory-, fatty-acid-oxidation-and-transport-, and mitochondrion-related complexes were downregulated in response to training AE in the YM but not in the elderly. The training volume may critically affect the changes in mitochondrial content, whereas exercise intensity may play a more important role in the changes of mitochondrial function [83]. Training AE contributes to profound adaptions of the cardiorespiratory system that enhance the delivery of oxygen from the atmosphere to the mitochondria and enable a tighter regulation of muscle metabolism [84]. Hence, compared to acute AE, a period of regular endurance training may have already reserved a wider adaptation in the function of metabolism and mitochondria. Notably, both acute AE and training AE were related to promoted skeletal muscle development, inflammation/immunity responses, and ECM remodeling.

Inconsistent findings were reported for the response of protein turnover and protein synthesis rate to AE combined with PI [9,37,45,85]. The types of protein synthesis influenced by varied modes of exercise and period of interventions may help explain the inconsistences [37]. Acute AE, but not training AE or its combination with PI, may have a more positive effect on regulating protein folding/refolding and contribute more in protein quality control, protein turnover, and protein synthesis [86]. Additionally, compared to training AE combined with PI, acute AE combined with PI may have a more obvious impact on reducing immune/inflammation response, ECM remodeling, the positive regulation of receptor signaling pathway via JAK-STAT and STAT, cellular response, and renal system vasculature development and protein turnover.

*UBB* encodes ubiquitin, which is involved in targeting cellular proteins for degradation by the 26S proteosome and the maintenance of chromatin structure, the regulation of gene expression, and the stress response. Previous studies suggested AE training prevents oxidative stress and ubiquitin-proteasome system overactivity and reverse skeletal muscle atrophy [87]. Exercises could counter these fibrotic changes by stimulating the activity of genes associated with ECM remodeling like *COL1A1*, which was related to ECM collagen synthesis [88]. As the hub gene of muscles, in response to training AE combined with PI, *MYC* was significantly upregulated in response to acute AE in the younger male but downregulated in AE combined with PI, which has a widespread impact on the transcriptome including regulation of cell growth arrest and adhesion, metabolism, ribosome biogenesis, protein synthesis, and mitochondrial function [89]. Moreover, *MYC* was suggested to be associated with insulin sensitivity [90], which might play a vital role in the muscle metabolism response to exercise and PI.

### 4.3. Strengths and Limitations

The strength of the present study was that this is the first study to compare the transcriptomic profiles of human skeletal muscle in response to AE, PI, and AE combined with PI. The findings provided the molecular evidence to support the positive role of PI in addition to acute AE, but limited molecular evidence was found for training AE combined with PI. This would help provide molecular hints for developing optimal measures to promote muscle health. The present study had some limitations. Firstly, there were heterogeneities among the five groups of intervention, such as the type and duration of AE, the dose and content of PI, which may had a profound impact on the transcriptomic changes and bias the comparisons. Secondly, demographic characteristics were not adjusted for in the comparison across varied interventions. Moreover, no blank control group was set in the studies of the intervention of AE combined with PI, which limited the power to assess the interaction effect of AE and PI.

## 5. Conclusions

The present study suggested that human skeletal muscle may respond to AE and PI in analogous molecular pathways but the responses may differ in direction. PI and additional PI based on acute AE may attenuate the immune/inflammation response and ECM remodeling which would be promoted by AE, irrespective of the type of AE, age and gender. Compared to PI alone, acute AE combined with PI would further promote protein turnover and synthesis, but may suppress skeletal muscle contraction and movement. The molecular impact on human skeletal muscle delivered by training AE combined with PI was not significant. Further studies exploring on the optimal dose of PI in combination with AE for synergic effects are needed.

## Figures and Tables

**Figure 1 nutrients-15-03485-f001:**
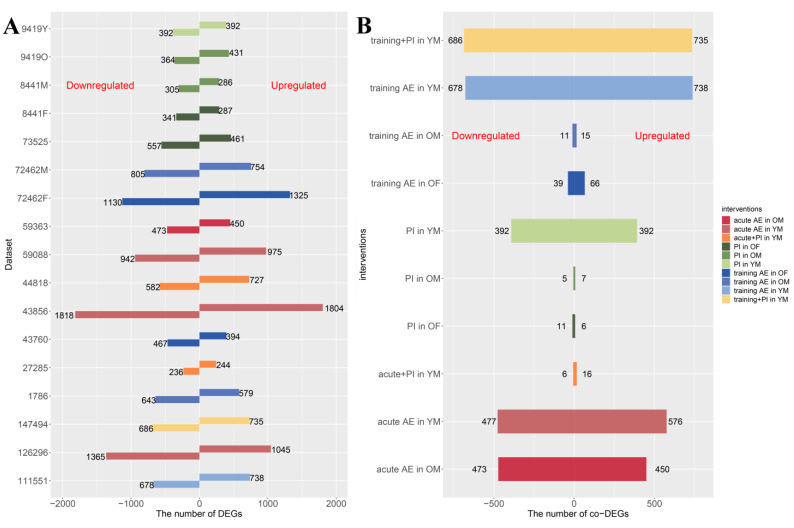
(**A**,**B**) The number of identified DEGs in each dataset and co-DEGs from each intervention among different populations. Note: AE, aerobic exercise; PI, protein intake; Y, the younger one; O, the older one; M, male; F, female; YM, the younger male; OM, the older male; OF, the older female.

**Figure 2 nutrients-15-03485-f002:**
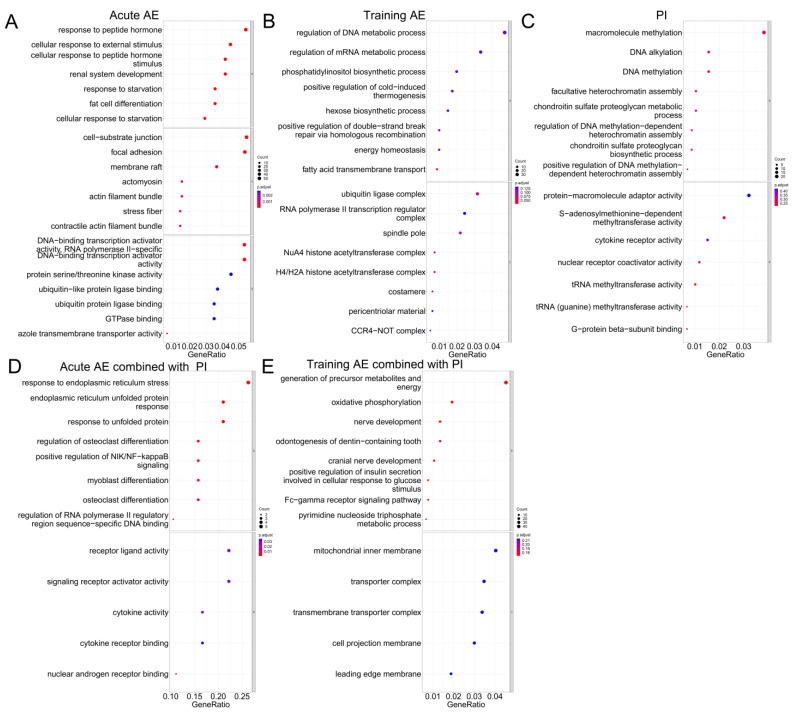
The GO enrichment analysis of co-DEGs from varied interventions in the younger male. (**A**) acute AE; (**B**) training AE; (**C**) PI; (**D**) acute AE combined with PI; (**E**) training AE combined with PI. Note: GO, Gene Ontology; co-DEGs, co-differentially expressed genes; AE, aerobic exercise; PI, protein intake.

**Figure 3 nutrients-15-03485-f003:**
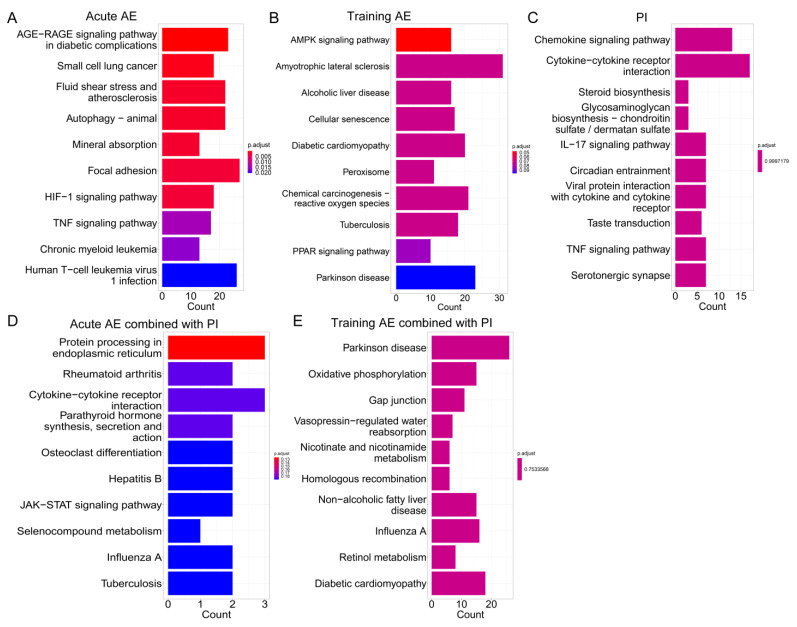
The KEGG enrichment analysis of co-DEGs from varied interventions in the younger male. (**A**) acute AE; (**B**) training AE; (**C**) PI; (**D**) acute AE combined with PI; (**E**) training AE combined with PI. Note: KEGG, Kyoto Encyclopedia of Genes and Genomes; co-DEGs, co-differentially expressed genes; AE, aerobic exercise; PI, protein intake.

**Figure 4 nutrients-15-03485-f004:**
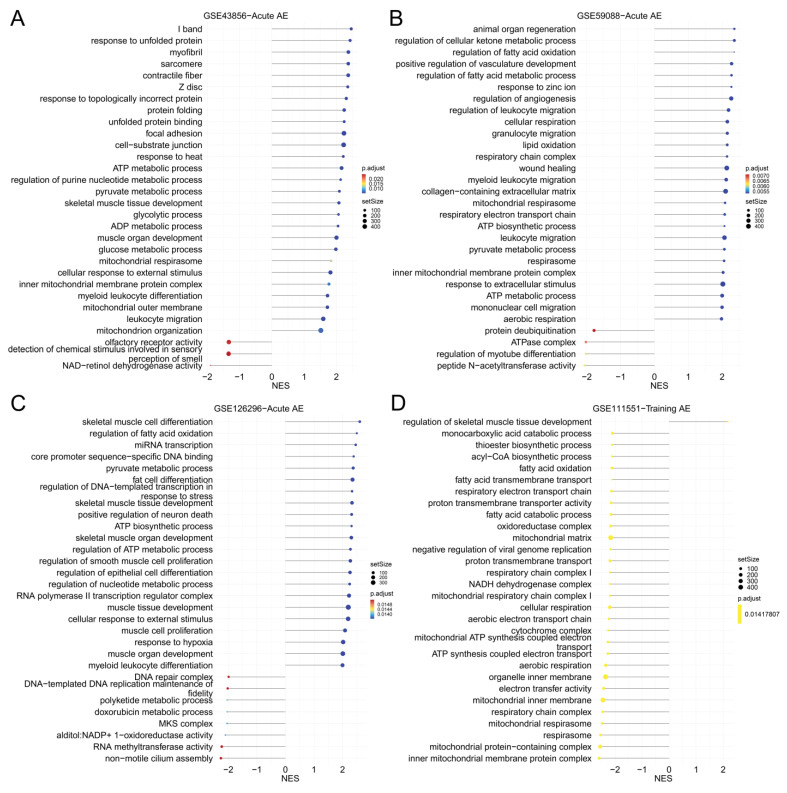
The gene sets related GO terms from GSEA in each dataset of AE in the younger male. (**A**) GSE43856—Acute AE; (**B**) GSE59088—acute AE; (**C**) GSE126296—acute AE; (**D**) GSE111551—training AE. Note: GO, Gene Ontology; GSEA, gene set enrichment analysis; AE, aerobic exercise; NES, normalized enrichment score.

**Figure 5 nutrients-15-03485-f005:**
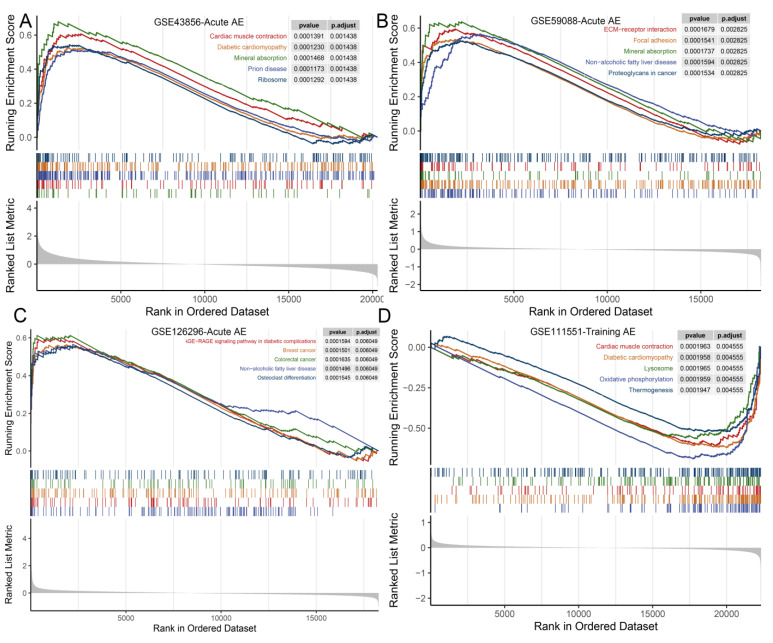
The top 5 KEGG pathway-related gene sets from GSEA in each dataset of AE in the younger male. (**A**) GSE43856—acute AE; (**B**) GSE59088—acute AE; (**C**) GSE126296—acute AE; (**D**) GSE111551—training AE. Note: KEGG, Kyoto Encyclopedia of Genes and Genomes; GSEA, gene set enrichment analysis; AE, aerobic exercise.

**Table 1 nutrients-15-03485-t001:** The characteristics of the studies included in the transcriptome analysis.

Datasets	Groups	Intervention Details	Age (Year)	Sex (n)	Muscle Biopsy	Platform
Acute aerobic exercise (n = 28, included 28 males)						
GSE43856 Neubauer O 2013, Austria [50]	Pre- and post-exercise	1 h of cycling followed by 1 h of running.	25 ± 4.1	M (8)	NR (3 h, 48 h, 96 h)	GPL10558; Illumina Human HT-12 V4.0 expression beadchip
GSE59088 Kristian Vissing 2014, Denmark [51]	Endurance group and control pre-, 2.5 h, 5 h	2 h of bicycle exercise at 60% post-training VO_2_ peak.	23.4 ± 0.8	M (6)	Vastus lateralis biopsy	GPL6244; [HuGene-1_0-st] Affymetrix Human Gene 1.0 ST Array [transcript (gene) version]
GSE126296 Rundqvist HC 2019, Sweden [52]	Pre- and post-exercise	Three all-out cycle sprints interspersed by 1/3-h recovery.	26 ± 4	M (7)	Vastus lateralis biopsy (7/3 h after the third sprint)	GPL6244; [HuGene-1_0-st] Affymetrix Human Gene 1.0 ST Array [transcript (gene) version]
GSE59363 Hansen JS 2015, Germany [53]	Pre- and post-exercise	A 1-h bicycle ergometer exercise	57.3 ± 1.7	M (7)	NR (0 h, 1 h, 3 h)	GPL6244; [HuGene-1_0-st] Affymetrix Human Gene 1.0 ST Array [transcript (gene) version]
Training aerobic exercise (n = 45, included 26 males and 19 females)						
GSE111551 Michelle Silva, 2018, Brazil	Pre- and post-training	18 weeks of running (3 times/week; 1 h every time).	Two groups group1, 26 ± 4; group2, 27 ± 3	M (13)	NR	GPL17586; [HTA-2_0] Affymetrix Human Transcriptome Array 2.0 [transcript (gene) version]
GSE72462 Böhm A, 2016, Germany [54]	Pre- and post-training	The training program lasted 8 weeks and consisted of three supervised exercise sessions per week. Each training session consisted of 0.5 h of bicycle ergometer exercise and 0.5 h walking on a treadmill.	46.4 ± 11	M (7) F (13)	Muscle biopsies were taken from the lateral portion of the vastus lateralis of the quadriceps femoris after local anesthesia	GPL17586; [HTA-2_0] Affymetrix Human Transcriptome Array 2.0 [transcript (gene) version]
GSE1786 Radom-Aizik S, 2005, USA [55]	Pre- and post-training	6 healthy sedentary men were trained on a cycle ergometer 3/wk for 12 wk at 80% of the predetermined maximal heart rate.	68 ± 2.7	M (6)	Vastus Lateralis muscle	GPL96; [HG-U133A] Affymetrix Human Genome U133A Array
GSE43760 Poelkens F, 2013, The Netherlands [56]	Pre- and post-training	6 months guided endurance exercise training program	49 ± 11	F (6)	Vastus lateralis muscle	GPL6244; [HuGene-1_0-st] Affymetrix Human Gene 1.0 ST Array [transcript (gene) version]
Protein intake (n = 48, included 27 males and 21 females)						
GSE8441 Thalacker-Mercer AE, 2007, USA [57]	Adequate protein vs. inadequate protein group	11 subjects consumed 1.2 g/kg/d protein for a week first, then consumed 0.5 g/kg/d protein for the second week.	55–80 (67 ± 7)	M (5) F (6)	Vastus lateralis biopsy	GPL96; [HG-U133A] Affymetrix Human Genome U133A Array
GSE9419 Thalacker-Mercer AE, 2008, USA [58]	Three groups with protein intakes of 0.50 g/kg/d, 0.75 g/kg/d, and 1.00 g/kg/d.	22 healthy young and old males treated with protein intakes of 0.50 g/kg/d, 0.75 g/kg/d, and 1.00 g/kg/d.	Young 21–43 Elderly 63–79	M (22)	Vastus lateralis biopsy	GPL570; [HG-U133_Plus_2] Affymetrix Human Genome U133 Plus 2.0 Array
GSE73525 Smith GI, 2016, Japan [59]	High-protein group vs. low-protein group	9 subjects in a high protein (1.2 g/kg/d) group and 6 subjects in a normal protein (0.8 g/kg/d) group	50–65 (58 ± 1)	F (15)	NR	GPL6244; [HuGene-1_0-st] Affymetrix Human Gene 1.0 ST Array [transcript (gene) version]
Acute aerobic exercise + protein intake (n = 20, included 20 males)						
GSE27285 Rowlands DS, 2011, Switzerland [60]	Acute aerobic exercise with or without protein intake (iso-calorie)	Group1: 1 h intense cycling + 0.2/1.2/0.4 g/kg (fat/carbohydrate/protein). Group2: 1 h intense cycling + 0.2/1.6 g/kg (fat/carbohydrate).	32.8 ± 6.4	M (8)	Quadriceps at rest, at 3 h and at 48 h after exercise training	GPL6104; Illumina humanRef-8 v2.0 expression beadchip
GSE44818 Rowlands DS, 2015, Switzerland [61]	High-intensity cycling without protein intake; high-intensity cycling + 70 g protein; high-intensity cycling + 23 g protein	5/3 h of high-intensity cycling with 70/15/180/30 g (protein/leucine/carbohydrate/fat); or with 23/5/180/30 g; or with 0/0/274/30 g	Mean age 30 years	M (12)	Vastus lateralis biopsy	GPL6947; Illumina HumanHT-12 V3.0 expression beadchip
Training aerobic exercise + protein intake (n = 40, included 40 males)						
GSE147494 Knuiman P, 2020, UK [62]	Protein-treated group vs. control group	12 weeks endurance training program with 30 g protein drink; or with 30 g carbohydrate drink (isocaloric)	18–30 Protein group 21.5; Controls group 22.5	M (40)	Vastus lateralis biopsy	GPL28236; [HuGene-2_1-st] Affymetrix Human Gene 2.1 ST Array

Notes: M, male; F, female; VO_2_, oxygen uptake per unit of time; GSE, array accession from the Gene Expression Omnibus; NR, not reported, GPL, Gene Platform. VO_2 peak_: peak oxygen uptake per unit of time.

## Data Availability

The summary-level datasets used and/or analyzed in the current study are available from the corresponding authors on reasonable request.

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
