# Peer review of "Transcriptomic Analysis of Human Skeletal Muscle in Response to Aerobic Exercise and Protein Intake"

_nutrients, 2023, doi:10.3390/nu15153485_

Round 1
Reviewer 1 Report
The authors aimed to highlight the divergent transcriptomic responses of human skeletal muscle to AE, PI and their combination, and to provide more molecular evidence of these interventions for maintaining muscle health and alleviating sarcopenia.
Line 17: was oppositely responded? Rephrase.
Line 17-19: too long sentence.
Line 39-43: What do you mean about physical performance? Strength performance is a branch of physical fitness. Keep strength performance and delete “physical”.
Line 52: Cardiovascular fitness
Line 53: what do you mean about oxidation capacity?
Line 53: And how about VO2max?
Line 59: Delete physical performance in the hole manuscript. Change it by “physical fitness”.
Line 376: “Compared to PI, acute AE combined with PI may have additional impact on upregulating protein turnover and synthesis and downregulating musculoskeletal movement and muscle contraction” and Line 414: “Meanwhile, the downregulation of skeletal muscle contraction, musculoskeletal movement, and muscle system development were found in response to acute AE combined with PI” …. A deeper discussion on this issue was missed.
It is fine, few edits needed, but it is ok.
Author Response
Authors' Response to Reviewers' Comments
Review 1
Comments and Suggestions for Authors
- The authors aimed to highlight the divergent transcriptomic responses of human skeletal muscle to AE, PI and their combination, and to provide more molecular evidence of these interventions for maintaining muscle health and alleviating sarcopenia.
Response: Sincerely appreciate for all your comments and suggestions, which help us prepare a better version. Please find the revised manuscript and the point-to-point responses as below for your review.
- Line 17: was oppositely responded? Rephrase.
- Line 17-19: too long sentence.
Response: Thank you for your suggestion. Please see the rephrased sentences in Line 17-19, Page 1, as “Both FOS and MYC were upregulated in response to acute AE, and they were downregulated by higher PI and the combination of AE and PI, respectively. COL1A1 was upregulated by training AE and was downregulated by higher PI.”
- Line 39-43: What do you mean about physical performance? Strength performance is a branch of physical fitness. Keep strength performance and delete “physical”.
Response: Physical performance here meant function performance, such as walking speed and balance. The term of physical has been deleted and function has been added. Please see the sentence in Line 42, Page 2.
- Line 52: Cardiovascular fitness
Response: Thank you. The term of “cardiovascular condition” has been changed into “cardiovascular fitness”. Please see the words in Line 54, Page 2.
- Line 53: what do you mean about oxidation capacity?
Response: Sorry that the term “oxidation capacity” in Line 50 refers to b-oxidation capacity but we forgot to add “b” [1]. Beta-oxidation capacity means that fatty acids, under the action of a series of enzymes α Carbon atoms and β break between carbon atoms to generate Acetyl-CoA and fatty acyl-coenzyme A. We have corrected this term to “b-oxidation capacity”. The second “oxidation capacity” in Line 54 in endurance exercise refers to the ability of resistance and/or aerobic exercise. To avoid misunderstanding, we have changed this term to “exercise capacity”.
[1] Kalinkovich, A.; Livshits, G. Sarcopenic obesity or obese sarcopenia: A cross talk between age-associated adipose tissue and skeletal muscle inflammation as a main mechanism of the pathogenesis. Ageing Res Rev 2017, 35, 200-221.
- Line 53: And how about VO2max?
Response: VO2max means peak oxygen uptake, which is expressed as the highest attained VO2 during the final 30s of exercise [2]. It refers to the amount of oxygen that can be absorbed when the body is unable to continue to support the next exercise when exercising at the maximum intensity. A higher VO₂ max represents a better exercise ability. It can be available by several measure methods such as laboratory measurement, indirect measurement, Bruce methods and so on.
[2] Beckers, P.J.; Denollet, J.; Possemiers, N.M.; Wuyts, F.L.; Vrints, C.J.; Conraads, V.M. Combined endurance-resistance training vs. endurance training in patients with chronic heart failure: a prospective randomized study. Eur Heart J 2008, 29, 1858-1866.
Line 59: Delete physical performance in the hole manuscript. Change it by “physical fitness”.
Response: Thank you for the suggestion. Please see the Response 4 above. Sorry for the misunderstanding brought by us. All the “physical performance” in the manuscript has been changed into “muscle function”.
- Line 376: “Compared to PI, acute AE combined with PI may have additional impact on upregulating protein turnover and synthesis and downregulating musculoskeletal movement and muscle contraction” and Line 414: “Meanwhile, the downregulation of skeletal muscle contraction, musculoskeletal movement, and muscle system development were found in response to acute AE combined with PI” …. A deeper discussion on this issue was missed.
Response: Thank you for your kind reminder. More discussion in relation to these two points have been added. Please see the revised Discussion on Page 19 to 20.
- Comments on the Quality of English Language
It is fine, few edits needed, but it is ok.
Response: Thank you. The grammar has been checked throughout the manuscript.
Reviewer 2 Report
1. Uniformize the dimension symbols in Table I: g/kg/day or g/kg/d, h or hours, 60min or 60 min
2. Include Table 2 and Figure 5 in Supplements instead of Text.
1. Uniformize the dimension symbols in Table I: g/kg/day or g/kg/d, h or hours, 60min or 60 min
2. Include Table 2 and Figure 5 in Supplements instead of Text.
Author Response
Review 2
Comments and Suggestions for Authors
- Uniformize the dimension symbols in Table I: g/kg/day or g/kg/d, h or hours, 60min or 60 min
Response: Thank you for your suggestion. Please see the revised Table 1 on Page 5.
- Include Table 2 and Figure 5 in Supplements instead of Text.
Response: Thank you for your suggestion. The original Table 2 and Figure 5 have been included in the Supplements. Please see the Supplementary files on Page 21-22.
- Comments on the Quality of English Language
Uniformize the dimension symbols in Table I: g/kg/day or g/kg/d, h or hours, 60min or 60 min
Include Table 2 and Figure 5 in Supplements instead of Text.
Response: The revisions have been made. Thank you.
Reviewer 3 Report
The authors wanted to provide more comprehensive molecular insights into the effects of acute / training aerobic exercise (AE), modified protein intake (PI), and AE combined with modified PI on transcriptomic profiles in human skeletal muscle. They used 14 published datasets from the Gene Expression Omnibus (GEO) database. Hub genes were identified in response to acute AE (ACTB, IL6), training AE (UBB, COL1A1), PI (EZH2), acute AE combined with PI (DDIT3), and training AE combined with modified PI (MYC), respectively.
The authors concluded that the human skeletal muscle may respond to AE and PI in analogous molecular pathways but the responses may differ in direction. PI acute AE + PI may attenuate the immune/inflammatory response and ECM remodeling which would be promoted by AE, independent of the type of AE, age and gender. Compared to PI alone, acute AE + PI could further promote protein turnover and synthesis, but may suppress skeletal muscle contraction and movement. The molecular impact on human skeletal muscle delivered by training AE + PI was not significant.
Studies like this are of course very important, also for daily clinical practice in order to prevent or treat muscle wasting. The reviewer acknowledges also the huge amount of work of the authors within this study. However, there are a number of shortcomings which should be clarified. First of all, the manuscript needs a comprehensive editing regarding grammar and style. The authors deal with a very complex issue. Therefore, the results should be described as simple as possible. Whenever possible, use the simple past. Description of the results should be better structured. My major concern, however, is that the five groups are very heterogeneous. In the exercise groups, type of exercise (running or cycling), work load, and duration (acute and training) differs significantly. The modification of the protein intake is done in different way. If you modify the protein content – what about the changes in carbohydrate and lipid content? In one study, leucine was added to the diet. Leucine itself is an important regulator of protein synthesis. There is just one data set in the training AE + PI group. It is also not clear if all the changes within the transcriptome are also translated into the proteome. All these points may have an impact on the results. The authors should discuss this and should be cautious regarding conclusions of the results.
Minor: The color code in Figure 1 should be improved. The terms used in Figure 2, 4 and 5 are not self-explaining. What is ATP or ADP metabolic process? What is pyruvate metabolic process? What is the difference between fatty acid oxidation and fatty acid catabolic process and between mitochondrial ATP synthesis-coupled electron transport and ATP synthesis-coupled electron transport? And many others … You should also avoid the term “respiratory chain”. This is no more used because it is not a chain. The correct term is “electron transport complex (ETC)”.
The manuscript needs a comprehensive editing regarding grammar and style. The authors deal with a very complex issue. Therefore, the results should be described as simple as possible. Whenever possible, use the simple past.
Author Response
Review 3
Comments and Suggestions for Authors
- The authors wanted to provide more comprehensive molecular insights into the effects of acute / training aerobic exercise (AE), modified protein intake (PI), and AE combined with modified PI on transcriptomic profiles in human skeletal muscle. They used 14 published datasets from the Gene Expression Omnibus (GEO) database. Hub genes were identified in response to acute AE (ACTB, IL6), training AE (UBB, COL1A1), PI (EZH2), acute AE combined with PI (DDIT3), and training AE combined with modified PI (MYC), respectively.
The authors concluded that the human skeletal muscle may respond to AE and PI in analogous molecular pathways but the responses may differ in direction. PI acute AE + PI may attenuate the immune/inflammatory response and ECM remodeling which would be promoted by AE, independent of the type of AE, age and gender. Compared to PI alone, acute AE + PI could further promote protein turnover and synthesis, but may suppress skeletal muscle contraction and movement. The molecular impact on human skeletal muscle delivered by training AE + PI was not significant.
Response: Thank you for all your comments and advice. Please find the revised manuscript and point-to-point responses as below for your reviewers.
- Studies like this are of course very important, also for daily clinical practice in order to prevent or treat muscle wasting. The reviewer acknowledges also the huge amount of work of the authors within this study. However, there are a number of shortcomings which should be clarified. First of all, the manuscript needs a comprehensive editing regarding grammar and style. The authors deal with a very complex issue. Therefore, the results should be described as simple as possible. Whenever possible, use the simple past. Description of the results should be better structured. My major concern, however, is that the five groups are very heterogeneous. In the exercise groups, type of exercise (running or cycling), work load, and duration (acute and training) differs significantly. The modification of the protein intake is done in different way. If you modify the protein content – what about the changes in carbohydrate and lipid content? In one study, leucine was added to the diet. Leucine itself is an important regulator of protein synthesis. There is just one data set in the training AE + PI group. It is also not clear if all the changes within the transcriptome are also translated into the proteome. All these points may have an impact on the results. The authors should discuss this and should be cautious regarding conclusions of the results.
Response: Sincerely appreciate for your comments and advice. The grammar and style have been checked and revised throughout the manuscript. The description of results has been edited. It is right that there was a relatively large heterogeneity among the intervention groups. However, the original studies available for the analyses limited the groups to be less heterogeneous. We have tried our best to group the original studies by their major characteristics. For example, time for exercise was an important criteria to categorize the groups for acute EE and training AE. In the acute AE group, all the subjects conducted 1-2h exercise, while in the training AE groups, all the subjects conducted at least 8 weeks training exercise. In addition, the dose and content used in the PI related original studies differed significantly. We have added the limitations accordingly in the section of Discussion. Please see the revised Page 21.
- Minor: The color code in Figure 1 should be improv The terms used in Figure 2, 4 and 5 are not self-explaining. What is ATP or ADP metabolic process? What is pyruvate metabolic process? What is the difference between fatty acid oxidation and fatty acid catabolic process and between mitochondrial ATP synthesis-coupled electron transport and ATP synthesis-coupled electron transport? And many others … You should also avoid the term “respiratory chain”. This is no more used because it is not a chain. The correct term is “electron transport complex (ETC)”.
Response: Thank you for all your kind reminders and suggestions. The color code in Figure 1 has been improved. Please see the revised Figure 1. Some terms labeled during the gene sets enrichment analysis (GSEA) and cited in Figure 2, 4 and 5 were derived from the original database. We didn’t use the term like “respiratory chain” in the text. Nevertheless, we have added the term“electron transport complex (ETC)”accordingly in the text.
- Comments on the Quality of English Language
The manuscript needs a comprehensive editing regarding grammar and style. The authors deal with a very complex issue. Therefore, the results should be described as simple as possible. Whenever possible, use the simple past.
Response: Thank you. The grammar and style have been checked and revised throughout the manuscript.
Round 2
Reviewer 1 Report
- Line 53: And how about VO2max?
Response: VO2max means peak oxygen uptake, which is expressed as the highest attained VO2 during the final 30s of exercise [2]. It refers to the amount of oxygen that can be absorbed when the body is unable to continue to support the next exercise when exercising at the maximum intensity. A higher VO₂ max represents a better exercise ability. It can be available by several measure methods such as laboratory measurement, indirect measurement, Bruce methods and so on.
[2] Beckers, P.J.; Denollet, J.; Possemiers, N.M.; Wuyts, F.L.; Vrints, C.J.; Conraads, V.M. Combined endurance-resistance training vs. endurance training in patients with chronic heart failure: a prospective randomized study. Eur Heart J 2008, 29, 1858-1866.
Be aware:Firstly,
"VO2max DOES NOT means peak oxygen uptake.
Unless there is demonstrable evidence that the plateau for a V̇O2max has been met in response to the exercise test
then the value maximally attained should be reported as the subject’s V̇O2peak .
See details here and edit your work regarding these issues:
10.1152/japplphysiol.00850.2017
10.1152/japplphysiol.00319.2018
Secondly,
"V̇" ... the dot should appear over the V to indicate "per unit of time")
"O2" for oxygen.
Thus, please insert, in the hole manuscritpt, the dot over the V for both V̇O2peak and V̇O2max.
It is fine!
Author Response
Comments and Suggestions for Authors
Reviewer 1
- Line 53: And how about VO2max?
1st Response: VO2max means peak oxygen uptake, which is expressed as the highest attained VO2 during the final 30s of exercise [2]. It refers to the amount of oxygen that can be absorbed when the body is unable to continue to support the next exercise when exercising at the maximum intensity. A higher VO₂ max represents a better exercise ability. It can be available by several measure methods such as laboratory measurement, indirect measurement, Bruce methods and so on.
[2] Beckers, P.J.; Denollet, J.; Possemiers, N.M.; Wuyts, F.L.; Vrints, C.J.; Conraads, V.M. Combined endurance-resistance training vs. endurance training in patients with chronic heart failure: a prospective randomized study. Eur Heart J 2008, 29, 1858-1866.
Be aware:
Firstly,
"VO2max DOES NOT means peak oxygen uptake.
Unless there is demonstrable evidence that the plateau for a V̇O2max has been met in response to the exercise test
then the value maximally attained should be reported as the subject’s V̇O2peak .
See details here and edit your work regarding these issues:
10.1152/japplphysiol.00850.2017
10.1152/japplphysiol.00319.2018
Secondly,
"V̇" ... the dot should appear over the V to indicate "per unit of time")
"O2" for oxygen.
Thus, please insert, in the hole manuscript, the dot over the V for both V̇O2peak and V̇O2max.
2nd Response: Sincerely appreciate for your correction. The revisions in relation to V̇O2peak and V̇O2max have been made. Please see Line 68, 158, 159 and the revised notes in Table 1.
Reviewer 3 Report
The authors considered most of my comment and suggestions and improved the manuscript accordingly.
However, the manuscript still needs an editing regarding grammar and style. Some sentences are simply too long and hard to understand.
However, the manuscript still needs an editing regarding grammar and style. Some sentences are not easy to understand or they are simply too long. The authors should contact a native speaker.
Author Response
Comments and Suggestions for Authors
Reviewer 3
The authors considered most of my comment and suggestions and improved the manuscript accordingly.
However, the manuscript still needs an editing regarding grammar and style. Some sentences are simply too long and hard to understand.
Comments on the Quality of English Language
However, the manuscript still needs an editing regarding grammar and style. Some sentences are not easy to understand or they are simply too long. The authors should contact a native speaker.
Response: Thank you for your suggestions. We have checked again and tried to polish the sentences, especially those in the Result section.